

SciPost Phys. 1(1), 009 (2016)

# A conformal bootstrap approach
# to critical percolation in two dimensions

## M. Picco[1], S. Ribault[2*] and R. Santachiara[3]

**1** LPTHE, CNRS and Université Pierre et Marie Curie, Sorbonne Universités
**2** Institut de physique théorique, CEA, CNRS, Université Paris-Saclay
**3** LPTMS, CNRS (UMR 8626), Université Paris-Saclay, 91405 Orsay, France

* sylvain.ribault@cea.fr

## Abstract

We study four-point functions of critical percolation in two dimensions, and more generally of the Potts model. We propose an exact ansatz for the spectrum: an infinite, discrete and non-diagonal combination of representations of the Virasoro algebra. Based on this ansatz, we compute four-point functions using a numerical conformal bootstrap approach. The results agree with Monte-Carlo computations of connectivities of random clusters.



# 1 Introduction

Critical percolation in two dimensions has a local conformal invariance [1–3], and can therefore be studied using the methods of two-dimensional conformal field theory. In particular, critical percolation has a Virasoro symmetry algebra with the central charge $c = 0$. While this particular value of the central charge gives rise to remarkable algebraic structures [4,5], one should also remember that critical percolation is the $q \to 1$ limit of the Potts model in the random cluster formulation [6], where the number $q$ is related to the central charge by

$$c = 1 - 6\left(\beta - \frac{1}{\beta}\right)^2, \quad q = 4\cos^2 \pi\beta^2 . \tag{1}$$

Some interesting observables of the Potts model are actually smooth as functions of $q$. These include the fractal dimensions of clusters and cluster boundaries [7], and the three-point connectivities of clusters [8]. So we will investigate critical percolation by studying smooth observables of the Potts model.

While three-point connectivities are interesting, the real test of one's understanding of the model comes with four-point connectivities. This is because conformal invariance reduces three-point functions to mere numbers, while four-point functions still depend non-trivially on a conformal cross-ratio. As a result, three-point functions cannot tell whether the involved fields are physical or not. In contrast, a four-point function encodes (part of) the spectrum of the model, and studying its limit when two points coincide can tell us what the ground state is, whether the spectrum is discrete or continuous, etc. The study of a boundary four-point function that obeys a second-order differential equation has led to Cardy's formula for the crossing probability in critical percolation [1]. Here we will study bulk four-point functions that do not obey any nontrivial differential equations.

Let us consider the random cluster formulation of the Potts model [6]. In this formulation, we draw graphs on a finite part of the square lattice $\mathbb{Z}^2$. A graph is a collection of bonds between neighbouring sites: each edge of the lattice either has a bond, or no bond. According to these bonds, the lattice is split into a disjoint union of connected clusters. The probability of a graph $\mathscr{G}$ is defined as

$$\text{Probability}(\mathscr{G}) = q^{\#\,\text{clusters}} p^{\#\,\text{bonds}} (1-p)^{\#\,\text{edges without bond}} . \tag{2}$$

The model becomes conformally invariant when the bond probability $p$ takes the critical value $p_c = \frac{\sqrt{q}}{\sqrt{q}+1}$ – the value at which the probability that there exists a percolating cluster jumps from 0 to 1, in the limit of infinite lattice size. Then the probability that two points $z_1, z_2$ belong to the same cluster behaves as [9]

$$P(z_1, z_2) \propto |z_1 - z_2|^{-4\Delta_{(0,\frac{1}{2})}} , \tag{3}$$

where the critical exponent $\Delta_{(0,\frac{1}{2})}$ is a function of $q$. This function is a special case of

$$\Delta_{(r,s)} = \frac{c-1}{24} + \frac{1}{4}\left(r\beta - \frac{s}{\beta}\right)^2 , \tag{4}$$

where the variables $c$ and $\beta$ are defined in terms of $q$ by eq. (1). The values of $\Delta_{(0,\frac{1}{2})}$ in a number of interesting cases, including critical percolation, are given in Table 1. The four basic four-point cluster connectivities that generalize $P(z_1, z_2)$ to configurations of four points

| Related model | $q$ | $c$ | $\Delta_{(0,\frac{1}{2})}$ |
|---|---|---|---|
| Uniform spanning tree | 0 | $-2$ | 0 |
| Critical percolation | 1 | 0 | $\frac{5}{96}$ |
| Ising model | 2 | $\frac{1}{2}$ | $\frac{1}{16}$ |
| Tricritical Ising model | $\frac{3+\sqrt{5}}{2}$ | $\frac{7}{10}$ | $\frac{21}{320}$ |
| Three-state Potts model | 3 | $\frac{4}{5}$ | $\frac{1}{15}$ |
| Tricritical three-state Potts model | $2+\sqrt{2}$ | $\frac{25}{28}$ | $\frac{15}{224}$ |
| Four-state Potts model | 4 | 1 | $\frac{1}{16}$ |

Table 1: Values of $q$, $c$, and $\Delta_{(0,\frac{1}{2})}$ in a number of special cases. The ambiguities in the relation (1) between $c$ and $q$ are lifted by assuming $\frac{1}{2} \leq \beta^2 \leq 1$.

$z_1, z_2, z_3, z_4$ are [10]

$$P_0(\{z_i\}) = \text{Probability}(z_1, z_2, z_3, z_4 \text{ are all in the same cluster}), \qquad (5)$$

$$P_1(\{z_i\}) = \text{Probability}(z_1, z_2 \text{ and } z_3, z_4 \text{ are in two different clusters}), \qquad (6)$$

$$P_2(\{z_i\}) = \text{Probability}(z_1, z_3 \text{ and } z_2, z_4 \text{ are in two different clusters}), \qquad (7)$$

$$P_3(\{z_i\}) = \text{Probability}(z_1, z_4 \text{ and } z_2, z_3 \text{ are in two different clusters}). \qquad (8)$$

The functions $P_1, P_2$ and $P_3$ are related to one another by permutations of their arguments,

$$P_1(z_1, z_2, z_3, z_4) = P_2(z_1, z_3, z_2, z_4) = P_3(z_1, z_3, z_4, z_2). \qquad (9)$$

Moreover, global conformal symmetry implies

$$P_\sigma\left(\left\{\frac{az_i + b}{cz_i + d}\right\}\right) = \prod_{i=1}^{4} |cz_i + d|^{4\Delta_{(0,\frac{1}{2})}} \cdot P_\sigma(\{z_i\}), \qquad (\sigma = 0, 1, 2, 3). \qquad (10)$$

Since the group of global conformal transformations $z \to \frac{az+b}{cz+d}$ is three-dimensional, it determines the dependence of $P_\sigma(\{z_i\})$ on only three of its four variables. The remaining fourth variable, which is invariant under these transformations, is the cross-ratio

$$z = \frac{(z_1 - z_2)(z_3 - z_4)}{(z_1 - z_3)(z_2 - z_4)}. \qquad (11)$$

This is why four-point functions encode much more information than two- and three-point functions.

We interpret the four-point connectivities $P_\sigma$ as four-point functions of conformal primary fields that all have dimensions $\Delta = \bar{\Delta} = \Delta_{(0,\frac{1}{2})}$. Assuming local conformal symmetry, such four-point functions are combinations of Virasoro conformal blocks $\mathscr{F}_\Delta^{(k)}(\{z_i\})$,

$$R = \sum_{(\Delta, \bar{\Delta}) \in \mathscr{S}^{(k)}} D_{\Delta, \bar{\Delta}}^{(k)} \mathscr{F}_\Delta^{(k)}(\{z_i\}) \mathscr{F}_{\bar{\Delta}}^{(k)}(\{\bar{z}_i\}), \qquad (k \in \{s, t, u\}). \qquad (12)$$

The index $k$ labels a channel, such that each formula for $R$ is an expansion around a given geometrical limit. (See Table 2.) Each term in the sum is the contribution of a primary state of left and right dimensions $\Delta$ and $\bar{\Delta}$, plus its descendent states. The equality of the expressions

| channel | limit |
|:---:|:---:|
| $s$ | $z_1 \to z_2$ |
| $t$ | $z_1 \to z_4$ |
| $u$ | $z_1 \to z_3$ |

Table 2: The three channels and the corresponding limits.

for $R$ in the $s$, $t$ and $u$ channels is a constraint on the spectrums $\mathscr{S}^{(k)}$ and on the structure constants $D^{(k)}_{\Delta,\bar{\Delta}}$, called crossing symmetry. The conformal bootstrap approach consists in solving this constraint. (Consistency of the theory on a torus would lead to the further constraint of modular invariance, which however applies to the complete spectrum of the theory, and does not constrain our OPE spectrums $\mathscr{S}^{(k)}$.)

In two-dimensional theories such as Virasoro minimal models, the spectrums are known, and finite. The crossing symmetry equations can then be solved exactly, resulting in analytic expressions for the structure constants [11]. On the other hand, in higher-dimensional theories such as the three-dimensional Ising model, only some qualitative features of the spectrums are known. Crossing symmetry can then be used for numerically estimating a few of the infinitely many dimensions $(\Delta, \bar{\Delta})$, and the associated structure constants [12,13]. Here we will follow the intermediate approach of numerically estimating a few structure constants, based on exact guesses for the spectrums.

## 2 Conformal bootstrap approach

### 2.1 Spectrums

What do we know on the spectrums $\mathscr{S}^{(k)}$ that should correspond to four-point functions such as the connectivities $P_\sigma$? First of all, in the limit $z_1 \to z_2$, the connectivity $P_0$ must reduce to the probability that $z_2, z_3, z_4$ are in the same cluster. It follows that the leading state of the corresponding spectrum, i.e. the state with the lowest total dimension $\Delta + \bar{\Delta}$, again has conformal dimensions $\Delta = \bar{\Delta} = \Delta_{(0,\frac{1}{2})}$. Moreover, conformal symmetry and single-valuedness of correlation functions only allow states with half-integer spins [11],

$$\Delta - \bar{\Delta} \in \frac{1}{2}\mathbb{Z} \, . \tag{13}$$

In particular, primary states such that $\Delta = \bar{\Delta}$ are called diagonal, and a spectrums where all primary states are diagonal is called diagonal too. Now, if we call "even spin" a spectrum where all primary states have even spins $\Delta - \bar{\Delta} \in 2\mathbb{Z}$, then

> a four-point function that has the same spectrum and structure constants in two channels, also has the same in the third channel, if and only if the spectrum is even spin.

(See Appendix A.2.) Our four-point function $P_0$ is invariant under permutations, and therefore has the same even spin spectrum in all channels. On the other hand, $P_1, P_2$ and $P_3$ are not invariant under all permutations, for instance $P_2$ should have the same spectrum and structure constants in the $s$- and $t$-channels, but not in the $u$-channel. Therefore, the $s$- and $t$-channel spectrum of $P_2$ cannot be even spin, and in particular cannot be diagonal.

Let us look for spectrums where all dimensions $\Delta, \bar{\Delta}$ are of the type $\Delta_{(r,s)}$ (4) with $(r,s) \in \mathbb{Z} \times \frac{1}{2}\mathbb{Z}$. For generic values of the central charge, the condition (13) then determines $\bar{\Delta}$ in

| Spectrum | Leading state | Even spin? |
|----------|---------------|------------|
| $\mathscr{S}_{2\mathbb{Z},\mathbb{Z}+\frac{1}{2}}$ | $(\Delta_{(0,\frac{1}{2})}, \Delta_{(0,\frac{1}{2})})$ | No |
| $\mathscr{S}_{2\mathbb{Z},\mathbb{Z}}$ | $(\Delta_{(0,0)}, \Delta_{(0,0)})$ | Yes |

Table 3: Two ansätze for the spectrums of four-point function.

terms of $\Delta$, namely $\Delta = \Delta_{(r,s)} \Rightarrow \bar{\Delta} = \Delta_{(r,-s)}$, so that $\Delta - \bar{\Delta} = -rs$. Our ansätze for spectrums will therefore be of the type

$$\mathscr{S}_{X,Y} = \left\{ (\Delta_{(r,s)}, \Delta_{(r,-s)}) \right\}_{r \in X, s \in Y} \qquad \text{with} \qquad X \subset \mathbb{Z}, \ Y \subset \frac{1}{2}\mathbb{Z} \,. \tag{14}$$

Spectrums of this type have been considered in [14], the most natural example being $\mathscr{S}_{\mathbb{Z},\mathbb{Z}}$. The total dimension of a state in such a spectrum is

$$\Delta_{(r,s)} + \Delta_{(r,-s)} = \frac{c-1}{12} + \frac{1}{2} \left( r^2 \beta^2 + \frac{s^2}{\beta^2} \right) \,. \tag{15}$$

We assume that the real part of the total dimension is bounded from below. Unless the sets $X$ and $Y$ are both finite, this implies $\Re \beta^2 > 0$, i.e.

$$\Re c < 13 \quad \Leftrightarrow \quad q \notin (4, \infty) \,. \tag{16}$$

The leading state of $\mathscr{S}_{X,Y}$, i.e. the state whose total dimension has the lowest real part, then has low values of $r, s$. We point out that there is no reason to assume that the values of $c$ or of conformal dimensions are real. Such assumptions would be necessary if we hoped to build unitary theories, but unitary theories cannot exist for generic values of $c < 1$ [11].

We are now ready to introduce our two main ansätze, see Table 3. Our main motivation for $\mathscr{S}_{2\mathbb{Z},\mathbb{Z}+\frac{1}{2}}$ is that it has the desired leading state. An additional motivation for both ansätze is that for $q = 4$, these spectrums are known to occur in four-point functions of the type of $P_\sigma$. Such four-point functions have indeed been computed in the Ashkin–Teller model, of which the four-state Potts model is a special case [15]. Moreover, the dimensions $\Delta_{(0,\mathbb{Z}+\frac{1}{2})}$ correspond to the magnetic series identified in [16–18], and the spectrum $\mathscr{S}_{2\mathbb{Z},\mathbb{Z}+\frac{1}{2}}$ appear in the partition functions discussed in [19].

## 2.2 Structure constants

Let us assume that we have the same known spectrum $\mathscr{S}^{(s)} = \mathscr{S}^{(t)} = \mathscr{S}$ in the $s$- and $t$-channels, with the same unknown structure constants $D^{(s)}_{\Delta,\bar{\Delta}} = D^{(t)}_{\Delta,\bar{\Delta}} = D_{\Delta,\bar{\Delta}}$. Let us determine these structure constants using the crossing symmetry equation

$$\sum_{(\Delta,\bar{\Delta}) \in \mathscr{S}} D_{\Delta,\bar{\Delta}} \left( \mathscr{F}^{(s)}_{\Delta}(\{z_i\}) \mathscr{F}^{(s)}_{\bar{\Delta}}(\{\bar{z}_i\}) - \mathscr{F}^{(t)}_{\Delta}(\{z_i\}) \mathscr{F}^{(t)}_{\bar{\Delta}}(\{\bar{z}_i\}) \right) = 0 \,. \tag{17}$$

This sum typically converges fast when the total dimension $\Delta + \bar{\Delta}$ increases. So let us truncate the spectrum, and consider the subspectrum $\mathscr{S}(N)$ made of the $N$ states with the lowest total dimensions. Normalizing the leading state structure constant to one, we determine the remaining $N-1$ structure constants of $\mathscr{S}(N)$ by randomly choosing $N-1$ values of the positions $\{z_i\}$. We call the spectrum consistent if the resulting structure constants are independent from the

| $(r, \quad s)$ | $(\Delta, \quad \bar{\Delta})$ | $D_{\Delta,\bar{\Delta}}(24)$ | $c_{\Delta,\bar{\Delta}}(24)$ |
|---|---|---|---|
| $\left(0, \quad \frac{1}{2}\right)$ | $\left(\frac{5}{96}, \quad \frac{5}{96}\right)$ | $1.0000000000$ | $0$ |
| $\left(-2, \quad \frac{1}{2}\right)$ | $\left(\frac{39}{32}, \quad \frac{7}{32}\right)$ | $0.0385548052$ | $1.3 \times 10^{-8}$ |
| $\left(2, \quad \frac{1}{2}\right)$ | $\left(\frac{7}{32}, \quad \frac{39}{32}\right)$ | $0.0385548052$ | $1.3 \times 10^{-8}$ |
| $\left(0, \quad \frac{3}{2}\right)$ | $\left(\frac{77}{96}, \quad \frac{77}{96}\right)$ | $-0.0212806512$ | $4.1 \times 10^{-8}$ |
| $\left(-2, \quad \frac{3}{2}\right)$ | $\left(\frac{95}{32}, \quad -\frac{1}{32}\right)$ | $0.0004525024$ | $1.2 \times 10^{-7}$ |
| $\left(2, \quad \frac{3}{2}\right)$ | $\left(-\frac{1}{32}, \quad \frac{95}{32}\right)$ | $0.0004525024$ | $1.2 \times 10^{-7}$ |
| $\left(0, \quad \frac{5}{2}\right)$ | $\left(\frac{221}{96}, \quad \frac{221}{96}\right)$ | $-0.0000356379$ | $2.5 \times 10^{-6}$ |
| $\left(-4, \quad \frac{1}{2}\right)$ | $\left(\frac{119}{32}, \quad \frac{55}{32}\right)$ | $-0.0000029746$ | $1.2 \times 10^{-5}$ |
| $\left(4, \quad \frac{1}{2}\right)$ | $\left(\frac{55}{32}, \quad \frac{119}{32}\right)$ | $-0.0000029746$ | $1.2 \times 10^{-5}$ |

Table 4: The first 9 states in the spectrum $\mathscr{S}_{2\mathbb{Z},\mathbb{Z}+\frac{1}{2}}$ at $c = 0$, together with the values of their conformal dimensions, of their structure constants, and of the coefficients of variation when the spectrum is truncated to $N = 24$ states. The coefficient of variation gives a rough estimate of the precision of our determination of the corresponding structure constant.

choice of $\{z_i\}$ in the limit $N \to \infty$. In practice, we randomly choose 10 values of $\{z_i\}$, and compute the mean $D_{\Delta,\bar{\Delta}}(N)$ and coefficient of variation $c_{\Delta,\bar{\Delta}}(N)$ of each structure constant. The structure constants $D_{\Delta,\bar{\Delta}}$ are then $D_{\Delta,\bar{\Delta}} = \lim_{N\to\infty} D_{\Delta,\bar{\Delta}}(N)$. For consistent spectrums, we find $c_{\Delta,\bar{\Delta}}(N \sim 20) < 10^{-5}$ for the first few structure constants. The precision of our conformal bootstrap has been evaluated also by testing it with the generalized minimal model correlation functions whose structure constants are known and given by Dotsenko-Fateev Coulomb gas integrals [20]. For inconsistent spectrums, we typically find $c_{\Delta,\bar{\Delta}}(N) > 10^{-2}$ for all $N$ and all structure constants.

## 2.3 Results

For all values of $c$ that obey (16), we find that the spectrum $\mathscr{S}_{2\mathbb{Z},\mathbb{Z}+\frac{1}{2}}$ is consistent. See Table 4 for the case $c = 0$. It takes a few minutes to compute these structure constants on a standard desktop computer. Then it takes a fraction of a second to compute the value of the four-point function at any given value of the positions $\{z_i\}$. See the corresponding Jupyter notebooks on GitHub for more details. And see Appendix B for numerical bootstrap results with other values of $N$ or of the number of choices of $\{z_i\}$.

The consistency of $\mathscr{S}_{2\mathbb{Z},\mathbb{Z}+\frac{1}{2}}$ allows us to define, and numerically compute, four-point functions that have the same symmetries as the four-point connectivities $P_1, P_2$ and $P_3$, and that we call $R_1, R_2$ and $R_3$. These four-point functions are related to one another by permutations of $\{z_i\}$. By definition, each one of these four-point functions has the spectrum $\mathscr{S}_{2\mathbb{Z},\mathbb{Z}+\frac{1}{2}}$ in two channels, while its spectrum $\mathscr{S}_0$ in the third channel is a priori unknown. (See Table 2.3.) In the case $c = 1$, we have $\mathscr{S}_0 = \mathscr{S}_{2\mathbb{Z},\mathbb{Z}}$ [15]. Generalizing this to $c \neq 1$ raises the issue that conformal blocks have poles at the values $\Delta = \Delta_{(r,s)}$ with $(r,s) \in 2\mathbb{N}^* \times \mathbb{N}^*$. As we explain in Appendix A.3, there is a natural regularization of these poles, at the price of allowing diagonal states with dimensions $\Delta = \bar{\Delta} = \Delta_{(r,-s)}$. With this regularization, we however find that the spectrum $\mathscr{S}_{2\mathbb{Z},\mathbb{Z}}$ is inconsistent. And this result seems independent from the regularization, whose influence is numerically rather small.

|  | $s$ | $t$ | $u$ |
|---|---|---|---|
| $R_1$ | $\mathscr{S}_0$ | $\mathscr{S}_{2\mathbb{Z},\mathbb{Z}+\frac{1}{2}}$ | $\mathscr{S}_{2\mathbb{Z},\mathbb{Z}+\frac{1}{2}}$ |
| $R_2$ | $\mathscr{S}_{2\mathbb{Z},\mathbb{Z}+\frac{1}{2}}$ | $\mathscr{S}_{2\mathbb{Z},\mathbb{Z}+\frac{1}{2}}$ | $\mathscr{S}_0$ |
| $R_3$ | $\mathscr{S}_{2\mathbb{Z},\mathbb{Z}+\frac{1}{2}}$ | $\mathscr{S}_0$ | $\mathscr{S}_{2\mathbb{Z},\mathbb{Z}+\frac{1}{2}}$ |

Table 5: Spectrums of $R_1, R_2$ and $R_3$ in each of the three channels.

## 3  Comparison with Monte-Carlo calculations

### 3.1  Monte-Carlo calculations

We study the Potts model on a square domain of $\mathbb{Z}^2$, with $L^2$ sites and periodic boundary conditions. Thanks to global conformal symmetry (10), we can restrict the four points to be of the type $(z_1, z_2, z_3, z_4) = (0, \ell, i\ell, \ell\frac{z-1}{z+i})$, where $z$ is the cross-ratio (11). Writing $z = \rho e^{i\theta}$, we want to study the dependence of four-point functions on $\rho$ at fixed $\theta$.

Fixing $\theta$ makes it impossible that all four points belong to the lattice, i.e. have integer coordinates $z_i \in \mathbb{Z} + i\mathbb{Z}$. Let us explain how we deal with this problem. We consider the 15 values of $\rho$ such that $\Re\frac{z-1}{z-i} \in \{\frac{1}{16}, \cdots, \frac{15}{16}\}$. Assuming $\ell$ to be a multiple of 16 then ensures that all our coordinates are integer, except $\Im z_4$. So we compute our four-point function at the two nearest integers $[\Im z_4]$ and $[\Im z_4] + 1$, and evaluate it at $\Im z_4$ by assuming that it behaves linearly as a function of $\Im z_4$.

We therefore obtain four-point functions that depend not only on $\{z_i\}$, but also on two extra geometric parameters $\ell$ and $L$. These extra dependences take the form

$$P_\sigma^{\ell,L}(\{z_i\}) = \ell^{-8\Delta_{(0,\frac{1}{2})}} P_\sigma(\{z_i\}) \left[ 1 + b_1\left(\frac{\ell}{L}\right)^\nu + b_2\left(\frac{\ell}{L}\right)^{2\nu} + \cdots \right] \times \left[ 1 + \frac{c_1}{\ell} + \frac{c_2}{\ell^2} + \cdots \right], \quad (18)$$

which involves small distance corrections as powers of $\frac{1}{\ell}$, and finite size corrections as powers of $\left(\frac{\ell}{L}\right)^\nu$, where $\nu = \frac{2}{3}\frac{\beta^2}{2\beta^2-1}$ is the correlation length exponent. Fitting our numerical results for $P_\sigma^{\ell,L}(\{z_i\})$ allows us to determine the coefficients $b_k, c_k$, and the sought after four-point function $P_\sigma(\{z_i\})$. The fits are done with a least-square Levenberg-Marquardt algorithm, using 47 values of $\ell$, and allowing coefficients $b_k, c_k$ for $k \leq 4$.

In practice, the probabilities $P_\sigma^{\ell,L}(\{z_i\})$ are evaluated on $N = 10^5$ independent configurations on a lattice of linear size $L = 8192$. (We checked that finite size corrections are negligible for this value of $L$, i.e. $|P(L = 8192) - P(L = 2048)|/P(L = 8192) < 10^{-3}$. For each configuration, we actually make $L^2$ measurements of $P_\sigma^{\ell,L}(\{z_i\})$, by varying the origin $z_1 = 0$ of our four-point configuration over the whole lattice. So $P_\sigma^{\ell,L}(\{z_i\})$ is actually an average over about $NL^2 \sim 10^{13}$ measurements. Testing how well the resulting four-point function $P_\sigma(\{z_i\})$ obeys global conformal symmety (10) allows us to estimate the relative error to $O(10^{-3})$.

In the case $q = 1$ of percolation, the Potts model is particularly easy to simulate, as the graph probability (2) does not depend on the number of clusters. Simulating the Potts model for $q \neq 1$ is more involved [21, 22]. In this article we present results for $1 \leq q \leq 3$.

### 3.2  Comparison

We find that $R_1, R_2, R_3$ are linear combinations of $P_0, P_1, P_2, P_3$ of the type

$$R_\sigma = \lambda(P_0 + \mu P_\sigma), \qquad (\sigma = 1, 2, 3), \quad (19)$$

| $q$ | $\lambda$ | $\delta\lambda$ | $\mu$ | $\delta\mu$ |
|------|------|------|------|------|
| 1.0 | 0.9563 | $2.8 \times 10^{-4}$ | $-2.0$ | 0.01 |
| 1.25 | 0.9426 | $1.7 \times 10^{-4}$ | $-3.32$ | 0.06 |
| 1.5 | 0.9281 | $3.9 \times 10^{-5}$ | $-5.95$ | 0.07 |
| 1.75 | 0.9142 | $2.7 \times 10^{-4}$ | $-13.85$ | 0.28 |
| 2.25 | 0.8881 | $2.5 \times 10^{-4}$ | $9.05$ | 0.48 |
| 2.5 | 0.8722 | $2.8 \times 10^{-4}$ | $4.46$ | 0.51 |
| 2.75 | 0.8555 | $6.1 \times 10^{-4}$ | $3.48$ | 0.76 |
| 3.0 | 0.8391 | $3.4 \times 10^{-4}$ | $2.0$ | 0.67 |

Table 6: Values of the coefficients $\lambda$ and $\mu$, together with the estimated errors, as functions of $q$.

where $\lambda$ and $\mu$ are $q$-dependent coefficients. (See Table 3.2.) The term $\mu P_\sigma$ is typically quite small compared to the term $P_0$, which explains why the error $\delta\mu$ is quite large. At $q = 2$, $\lambda$ is smooth but $\mu$ diverges. This divergence is an artefact of our normalization assumption $D_{\Delta_{(0,\frac{1}{2})},\Delta_{(0,\frac{1}{2})}} = 1$: this structure constant actually goes to zero relative to other structure constants in the limit $q \to 2$.

With these values of $\lambda$ and $\mu$, let us plot the values of $R_3$ from the conformal bootstrap analysis of Section 2 (with dots), and from Monte-Carlo calculations of the right-hand side of (19) (with crosses). Using global conformal symmetry, we can reduce $R_3(\{z_i\})$ to a function of the cross-ratio $z = \rho e^{i\theta}$, that we plot in Figure 1. We find a good agreement between Monte-Carlo and conformal bootstrap results, with small errors. (See Figure 3.2.) The $\lambda$ and $\mu$ parameters are determined by imposing that the difference remains close to zero for all values of $\rho$.

### 3.3 Interpretation

Inspired by the known results at $c = 1$ [15], let us propose a tentative interpretation of the functions $P_\sigma, R_\sigma$ as four-point functions of conformal fields.

Let $V_+, V_-$ be two primary fields with the same dimensions $\Delta = \bar\Delta = \Delta_{(0,\frac{1}{2})}$, that are related by a $\mathbb{Z}_2$ symmetry. The spectrum $\mathscr{S}_{2\mathbb{Z},\mathbb{Z}+\frac{1}{2}}$ controls the operator product expansion of $V_-V_+$, while the unknown spectrum $\mathscr{S}_0$ controls $V_-V_-$ and $V_+V_+$. The two-point functions are $\langle V_+V_- \rangle = 0$ and $\langle V_-(z_1)V_-(z_2) \rangle = \langle V_+(z_1)V_+(z_2) \rangle \propto P(z_1, z_2)$, where $P(z_1, z_2)$ is the two-point connectivity (3). Correspondingly, a state with dimensions $\Delta = \bar\Delta = 0$ should appear in $\mathscr{S}_0$, whereas there is no such state in $\mathscr{S}_{2\mathbb{Z},\mathbb{Z}+\frac{1}{2}}$. The functions $R_1, R_2$ and $R_3$ are the four-point functions

$$R_1 = \langle V_-V_-V_+V_+ \rangle = \langle V_+V_+V_-V_- \rangle \,, \tag{20}$$

$$R_2 = \langle V_-V_+V_-V_+ \rangle = \langle V_+V_-V_+V_- \rangle \,, \tag{21}$$

$$R_3 = \langle V_-V_+V_+V_- \rangle = \langle V_+V_-V_-V_+ \rangle \,. \tag{22}$$

Four-point functions such as $\langle V_-V_-V_-V_+ \rangle$ vanish. We define

$$R_0 = \langle V_-V_-V_-V_- \rangle = \langle V_+V_+V_+V_+ \rangle \,, \tag{23}$$

a permutation-symmetric four-point function whose spectrum is $\mathscr{S}_0$ in all channels, such that

$$R_0 = \lambda \left( P_0 + \mu P_1 + \mu P_2 + \mu P_3 \right) \,. \tag{24}$$

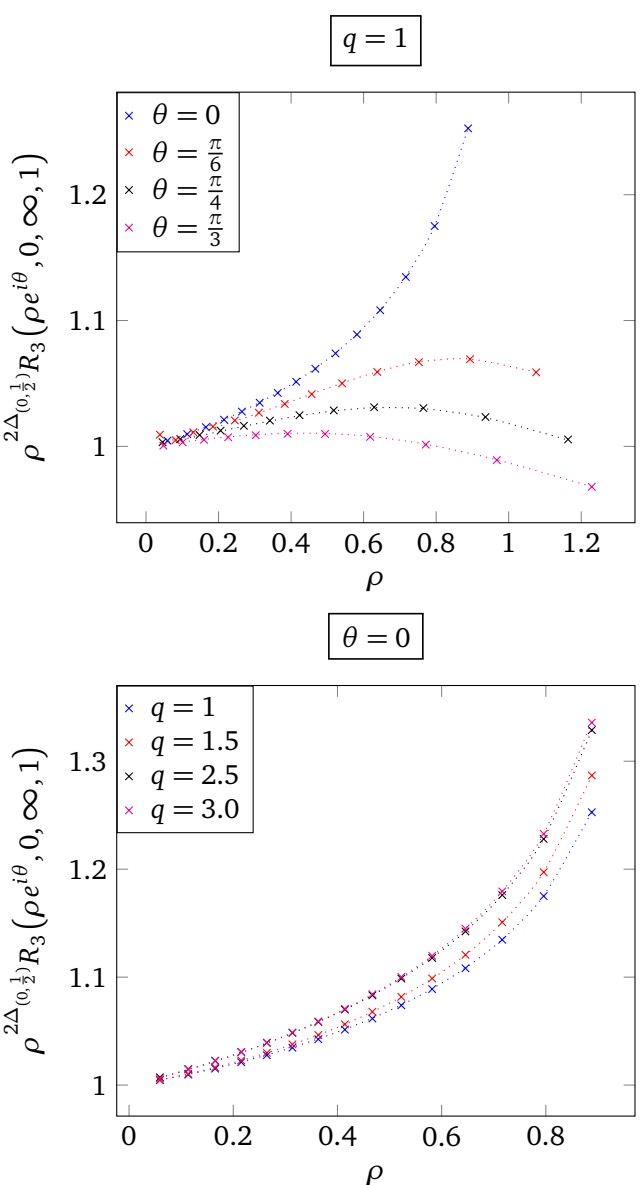

Figure 1: $R_3$ vs. $\rho$ for $c = 0$ and $\theta = 0, \pi/6, \pi/4, \pi/3$, and for $\theta = 0$ and $q \in \{1, 1.5, 2.5, 3\}$. Crosses correspond to the Monte-Carlo simulations. The dotted lines show the bootstrap solution, obtained by truncating the spectrum at $N = 6$ states and the expansion of the conformal blocks at level $L = 10$.

SciPost Phys. 1(1), 009 (2016)

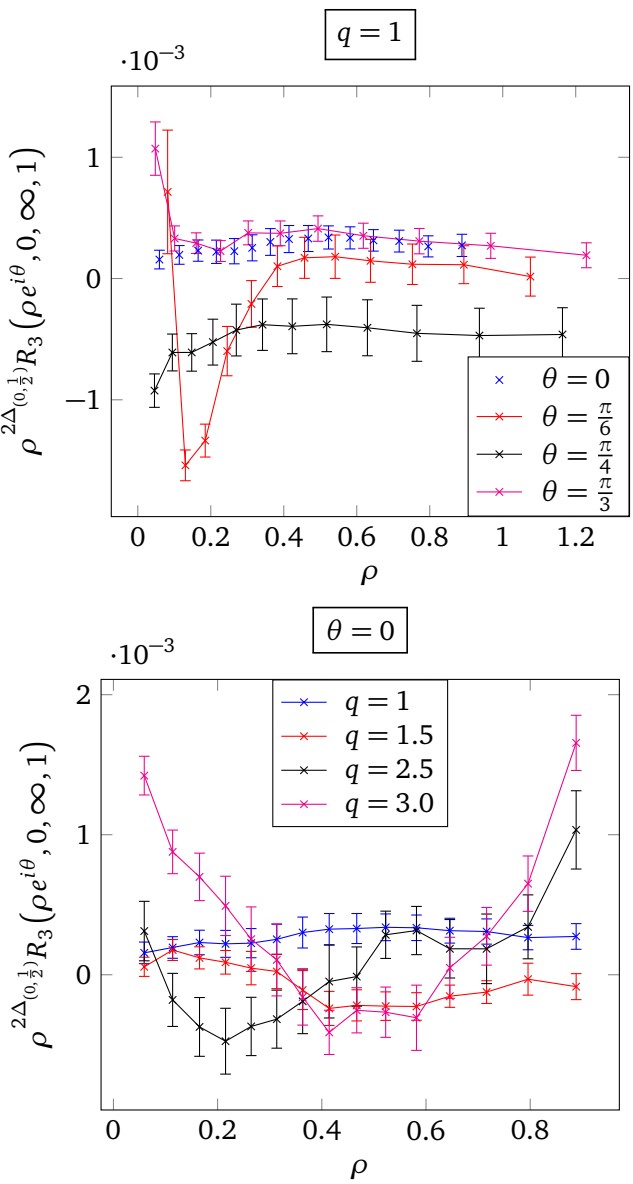

Figure 2: Differences between Monte-Carlo and numerical bootstrap calculations, with the error bars of the Monte-Carlo simulations. In all cases the difference is of the order of $10^{-3}$ or less, while the results themselves are of order 1.

| $R$ | $P_0$ | $P_1$ | $P_2$ | $P_3$ | $R_0$ | $R_1$ | $R_2$ | $R_3$ | |
|---|---|---|---|---|---|---|---|---|---|
| $\Delta$ | $\Delta_{(0,\frac{1}{2})}$ | $\leq 0$ | $> \Delta_{(0,\frac{1}{2})}$ | $> \Delta_{(0,\frac{1}{2})}$ | $\leq 0$ | $\leq 0$ | $\Delta_{(0,\frac{1}{2})}$ | $\Delta_{(0,\frac{1}{2})}$ | (27) |

Table 7: Critical exponents $\Delta$ of $P_\sigma$ and $R_\sigma$ in the limit $z_1 \to z_2$, such that these quantities behave as $|z_1 - z_2|^{2\Delta - 4\Delta_{(0,\frac{1}{2})}}$.

We can then write expressions for $P_\sigma$ in terms of the fields $V_\pm$, in particular

$$P_0 = \frac{1}{4\lambda}\Big\langle (V_- + iV_+)(V_- + iV_+)(V_- + iV_+)(V_- + iV_+) \Big\rangle , \tag{25}$$

$$P_1 = \frac{1}{4\lambda\mu}\Big\langle (V_- + V_+)(V_- + V_+)(V_- - V_+)(V_- - V_+) \Big\rangle . \tag{26}$$

Our interpretation is consistent with the behaviour of $P_\sigma$ and $R_\sigma$ in the limit $z_2 \to z_1$ for $0 \leq c \leq 1$. (See Table 7.) We infer the behaviour of $P_\sigma$ from our Monte-Carlo results, the behaviour of $R_1, R_2, R_3$ from our conformal bootstrap results, and the behaviour of $R_0$ from the observation that it involves the same operator product $V_- V_-$ as $R_1$.

Notice that the fields $V_\pm$ are strongly reminiscent of the couple of magnetic fields discussed in [8], which were argued to be at the origin of a factor $\sqrt{2}$ in three-point connectivities [8,23].

# 4 Conclusion

Our conformal bootstrap results allow us to compute three linear combinations $R_1, R_2, R_3$ of the four connectivities $P_\sigma$. Determining the missing combination amounts to finding the spectrum $\mathscr{S}_0$ of $R_1$ in the $s$-channel. Our guesses for this spectrum, including the expression $\mathscr{S}_0 = \mathscr{S}_{2\mathbb{Z},\mathbb{Z}}$ that is valid at $c = 1$, have so far been wrong. Since $R_1$ can be computed with very good accuracy, it should however be possible to determine $\mathscr{S}_0$ numerically. In particular it would be interesting to find out whether the leading state's total conformal dimension is 0, as expected on general statistical physics ground, or $2\Delta_{(0,0)} = \frac{c-1}{12}$, as in $\mathscr{S}_{2\mathbb{Z},\mathbb{Z}}$, or something else.

From the four-point functions $R_1$, $R_2$ and $R_3$, it is possible to deduce three-point structure constants, whose interpretation in the Potts model remains to be found. Three-point connectivities of clusters are known to be related to structure constants of the Liouville conformal field theory with $c \leq 1$ [8, 23–25], but that theory has a continuous, diagonal spectrum [26] which is very different from the discrete, non-diagonal spectrum $\mathscr{S}_{2\mathbb{Z},\mathbb{Z}+\frac{1}{2}}$ that we found. Nevertheless, both spectrums have an important feature in common: they are built from Verma modules of the Virasoro algebra, that is from representations where the Virasoro generator $L_0$ is diagonalizable. Our four-point functions therefore join the three-point connectivities on the list of non-trivial observables in bulk critical percolation that have no logarithmic features. This contrasts with other observables that involve Virasoro modules where $L_0$ is not diagonalizable [5, 27, 28].

The four-point functions that we computed are defined for any complex values of the number of states $q$, although the relation between (1) between $c$ and $q$ is ambiguous. The complicated dependence on $q$ opens the possibility of nontrivial phenomenons, including special behaviour at $q = 2$ and $q = 3$ where the Potts model is related to Virasoro and $W_3$ minimal models respectively [23], and the duality $\beta \to \frac{1}{\beta}$ of conformal field theory.

In the case $c = 0$, it was very recently proposed that a four-point function of fields of dimension $\Delta_{(0,\frac{1}{2})}$ has a Coulomb gas integral representation [29]. The corresponding spectrum

would contain only one diagonal field of dimension $\Delta_{(2,0)} = \frac{5}{8}$, and the four-point function would be symmetric under permutations. Such a four-point function however cannot be a linear combination of the connectivities $P_\sigma$, given what we know of their asymptotic behaviour (7).

We have investigated certain four-point functions of fields with dimensions $\Delta = \bar{\Delta} = \Delta_{(0,\frac{1}{2})}$, but our methods could be generalized to other four-point functions. To begin with, there exist several different interesting fields with these dimensions, including our fields $V_+$ and $V_-$, and the leading field in their operator product expansion. The case $c = 1$ suggests that there are more [15]. Moreover, we found that the spectrum $\mathscr{S}_{2\mathbb{Z}+1,\mathbb{Z}}$ is consistent, which suggests the existence of yet more fields of this type. And of course, nothing prevents the investigation of fields with other dimensions.

# Acknowledgements

We are grateful to Gesualdo Delfino, Vladimir Dotsenko, Sheer El-Showk, Benoît Estienne, Yacine Ikhlef, Jesper Jacobsen, Santiago Migliaccio, Miguel Paulos, Hubert Saleur, and Jacopo Viti, for useful discussions and comments. We wish to thank the referees for their work, and SciPost for publishing their reports.

# A   Conformal blocks

In this Appendix we collect some useful properties of Virasoro conformal blocks. The conformal blocks that we need appear in four-point functions of fields with conformal dimension $\Delta_{(0,\frac{1}{2})}$. There are some simplifications in this case, but the properties that we will discuss can be generalized to four fields with arbitrary dimensions. See the review article [11] for more explanations on Virasoro conformal blocks.

Using global conformal symmetry, we can set $(z_1, z_2, z_3, z_4) = (z, 0, \infty, 1)$, and we call $\mathscr{F}_\Delta^{(k)}(z) = \mathscr{F}_\Delta^{(k)}(z, 0, \infty, 1)$ the resulting conformal blocks.

## A.1   Zamolodchikov's recursive formula

Conformal blocks can be numerically computed using the formula

$$\mathscr{F}_\Delta^{(s)}(z) = (16q)^{\Delta - \frac{c-1}{24}} \left(z(1-z)\right)^{-\frac{c-1}{24} - \frac{1}{8\beta^2}} \theta_3(q)^{-\frac{c-1}{6} - \frac{1}{\beta^2}} H_\Delta(q) \,, \tag{28}$$

where the elliptic nome $q$ and function $\theta_3(q)$ are given by

$$q = \exp -\pi \frac{F(\frac{1}{2}, \frac{1}{2}, 1, 1-z)}{F(\frac{1}{2}, \frac{1}{2}, 1, z)}, \qquad \theta_3(q) = \sum_{n \in \mathbb{Z}} q^{n^2} \,, \tag{29}$$

and the factor $H_\Delta(q)$ obeys the recursion relation

$$H_\Delta(q) = 1 + \sum_{r,s=1}^\infty \frac{(16q)^{rs}}{\Delta - \Delta_{(r,s)}} R_{r,s} H_{\Delta_{(r,-s)}}(q) \,. \tag{30}$$

The coefficients $R_{r,s}$ are defined by

$$R_{r,s} \underset{r \text{ odd}}{=} 0 \,, \tag{31}$$

$$R_{r,s} \underset{r \text{ even}}{=} -2^{1-4rs} P_{(0,0)} P_{(r,s)} \prod_{r'=1-r}^{r} \prod_{s'=1-s}^{s} P_{(r',s')}^{(-1)^{r'+1}} \,, \tag{32}$$

where $P_{(r,s)} = \frac{1}{2}\left(r\beta - \frac{s}{\beta}\right)$, and the factor $P_{(0,0)} = 0$ is actually meant to cancel with the same factor from the denominator. Notice that for $c = 1$ we have $R_{r,s} = 0$ and $H_\Delta(q) = 1$.

The recursion relation determines $H_\Delta(q)$ by giving its poles and residues as a function of $\Delta$. In general four-point blocks, poles appear when $\Delta$ take the values $\Delta = \Delta_{(r,s)}$ with $r, s \in \mathbb{N}^*$, that correspond to reducible Verma modules. However, in our particular four-point blocks, poles with odd $r$ have vanishing residues.

## A.2 Crossing symmetry and even spin spectrums

Let us justify the property of even spin spectrums that is invoked in Section 2.1. This discussion is inspired from Section 7 of [30]. To begin with, let us write $t$- and $u$-channel conformal blocks in terms of $s$-channel conformal blocks. The different channels are related by permutations of $\{z_i\}$, and this implies

$$\mathcal{F}_\Delta^{(t)}(z) = \mathcal{F}_\Delta^{(s)}(1-z), \qquad \mathcal{F}_\Delta^{(u)}(z) = z^{-2\Delta}\mathcal{F}_\Delta^{(s)}(\tfrac{1}{z}) . \tag{33}$$

Assuming that our spectrum and structure constants obey the $s-t$ crossing symmetry equation (17), the agreement with the $u$-channel becomes equivalent to

$$\sum_{(\Delta,\bar{\Delta})\in\mathscr{S}} D_{\Delta,\bar{\Delta}}\left( \mathcal{F}_\Delta^{(s)}(z)\mathcal{F}_{\bar{\Delta}}^{(s)}(\bar{z}) - |z-1|^{-4\Delta_{(0,\frac{1}{2})}}\mathcal{F}_\Delta^{(s)}(\tfrac{z}{z-1})\mathcal{F}_{\bar{\Delta}}^{(s)}(\tfrac{\bar{z}}{\bar{z}-1}) \right) = 0 . \tag{34}$$

Using the identities

$$q(\tfrac{z}{z-1}) = -q, \qquad \theta_3(-q) = (z-1)^{\frac{1}{4}}\theta_3(q), \qquad H_\Delta(-q) = H_\Delta(q) , \tag{35}$$

the agreement with the $u$-channel becomes

$$\sum_{(\Delta,\bar{\Delta})\in\mathscr{S}} D_{\Delta,\bar{\Delta}}\left(1-(-1)^{\Delta-\bar{\Delta}}\right)\mathcal{F}_\Delta^{(s)}(z)\mathcal{F}_{\bar{\Delta}}^{(s)}(\bar{z}) = 0 . \tag{36}$$

This vanishes if and only if all spins $\Delta - \bar{\Delta}$ in the spectrum $\mathscr{S}$ are even. Therefore, this even spin condition is necessary and sufficient for our four-point function to be symmetric under all permutations of $\{z_i\}$, in other words to have the same spectrum and structure constants in the $u$-channel as in the $s$- and $t$-channels.

## A.3 Logarithmic regularization

In order to regularize a conformal block at its pole $\Delta = \Delta_{(r,s)}$, we might be tempted to take the residue,

$$\operatorname*{Res}_{\Delta=\Delta_{(r,s)}} \mathcal{F}_\Delta^{(s)}(z) = R_{r,s}\mathcal{F}_{\Delta_{(r,-s)}}^{(s)}(z) . \tag{37}$$

However, the resulting conformal block would behave as $O(z^{\Delta_{(r,-s)}-2\Delta_{(0,\frac{1}{2})}})$ near $z=0$, whereas we are looking for a regularization that behaves as $O(z^{\Delta_{(r,s)}-2\Delta_{(0,\frac{1}{2})}})$. So we multiply the block with the factor $\Delta - \Delta_{(r,s)}$ and then send $\Delta$ not to $\Delta_{(r,s)}$, but to $\Delta_{(r,s)} + \left(\begin{smallmatrix} 0 & 1 \\ 0 & 0 \end{smallmatrix}\right)$. The elements of the resulting matrix include not only $\operatorname*{Res}_{\Delta=\Delta_{(r,s)}} \mathcal{F}_\Delta^{(s)}(z)$, but also the regularized block that would be obtained by using the recipe

$$\lim_{\Delta\to\Delta_{(r,s)}} \frac{1}{\Delta - \Delta_{(r,s)}} = \log(16q) , \tag{38}$$

in eq. (30). Using this regularization implies that we must also allow a contribution of $\mathcal{F}_{\Delta_{(r,-s)}}^{(s)}(z)$ with an unknown coefficient.

This regularization has an algebraic interpretation in terms of representations of the Virasoro algebra where the Virasoro generator $L_0$ is not diagonalizable.

| $(r,\ s)$ | $(\Delta,\ \bar{\Delta})$ | $D_{\Delta,\bar{\Delta}}(6)$ | $c_{\Delta,\bar{\Delta}}(6)$ |
|---|---|---|---|
| $\left(0,\ \frac{1}{2}\right)$ | $\left(\frac{5}{96},\ \frac{5}{96}\right)$ | 1.0000000000 | 0 |
| $\left(-2,\ \frac{1}{2}\right)$ | $\left(\frac{39}{32},\ \frac{7}{32}\right)$ | 0.0391621873 | 0.0123 |
| $\left(2,\ \frac{1}{2}\right)$ | $\left(\frac{7}{32},\ \frac{39}{32}\right)$ | 0.0391621873 | 0.0123 |
| $\left(0,\ \frac{3}{2}\right)$ | $\left(\frac{77}{96},\ \frac{77}{96}\right)$ | $-0.0223661625$ | 0.0377 |
| $\left(-2,\ \frac{3}{2}\right)$ | $\left(\frac{95}{32},\ -\frac{1}{32}\right)$ | 0.0004107222 | 0.0979 |
| $\left(2,\ \frac{3}{2}\right)$ | $\left(-\frac{1}{32},\ \frac{95}{32}\right)$ | 0.0004107222 | 0.0979 |

Table 8: Truncating the spectrum $\mathscr{S}_{2\mathbb{Z},\mathbb{Z}+\frac{1}{2}}$ to $N=6$ states at $c=0$.

| $(r,\ s)$ | $(\Delta,\ \bar{\Delta})$ | $D_{\Delta,\bar{\Delta}}(13)$ | $c_{\Delta,\bar{\Delta}}(13)$ |
|---|---|---|---|
| $\left(0,\ \frac{1}{2}\right)$ | $\left(\frac{5}{96},\ \frac{5}{96}\right)$ | 1.0000000000 | 0 |
| $\left(-2,\ \frac{1}{2}\right)$ | $\left(\frac{39}{32},\ \frac{7}{32}\right)$ | 0.0385548455 | $5 \times 10^{-7}$ |
| $\left(2,\ \frac{1}{2}\right)$ | $\left(\frac{7}{32},\ \frac{39}{32}\right)$ | 0.0385548455 | $5 \times 10^{-7}$ |
| $\left(0,\ \frac{3}{2}\right)$ | $\left(\frac{77}{96},\ \frac{77}{96}\right)$ | $-0.0212807204$ | $1.7 \times 10^{-6}$ |
| $\left(-2,\ \frac{3}{2}\right)$ | $\left(\frac{95}{32},\ -\frac{1}{32}\right)$ | 0.000452499 | $3.7 \times 10^{-6}$ |
| $\left(2,\ \frac{3}{2}\right)$ | $\left(-\frac{1}{32},\ \frac{95}{32}\right)$ | 0.000452499 | $3.7 \times 10^{-6}$ |
| $\left(0,\ \frac{5}{2}\right)$ | $\left(\frac{221}{96},\ \frac{221}{96}\right)$ | $-0.0000356329$ | $9.8 \times 10^{-5}$ |
| $\left(-4,\ \frac{1}{2}\right)$ | $\left(\frac{119}{32},\ \frac{55}{32}\right)$ | $-0.0000029756$ | $6 \times 10^{-4}$ |
| $\left(4,\ \frac{1}{2}\right)$ | $\left(\frac{55}{32},\ \frac{119}{32}\right)$ | $-0.0000029756$ | $6 \times 10^{-4}$ |

Table 9: Truncating the spectrum $\mathscr{S}_{2\mathbb{Z},\mathbb{Z}+\frac{1}{2}}$ to $N=13$ states at $c=0$, while showing the results for the first 9 states only.

## B   More numerical conformal boostrap results

In order to show that our numerical conformal bootstrap method converges towards the announced results, let us vary $N$ and the number of choices of positions $\{z_i\}$, in the case of the spectrum $\mathscr{S}_{2\mathbb{Z},\mathbb{Z}+\frac{1}{2}}$ at the central charge $c=0$. The cases $N=6$ (Table 8), $N=13$ (Table 9) and $N=24$ (Table 4) show that given state, the coefficient of variation quickly decreases as $N$ increases. On the other hand, varying the number of choices of the points $\{z_i\}$ does not significantly change the coefficients of variation, see Table 4 (10 choices) and Table 10 (20 choices).

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

| $(r, \quad s)$ | $(\Delta, \quad \bar\Delta)$ | $D_{\Delta,\bar\Delta}(24)$ | $c_{\Delta,\bar\Delta}(24)$ |
|---|---|---|---|
| $\left(0, \tfrac{1}{2}\right)$ | $\left(\tfrac{5}{96}, \tfrac{5}{96}\right)$ | 1.0000000000 | 0 |
| $\left(-2, \tfrac{1}{2}\right)$ | $\left(\tfrac{39}{32}, \tfrac{7}{32}\right)$ | 0.038554810 | $2.6 \times 10^{-8}$ |
| $\left(2, \tfrac{1}{2}\right)$ | $\left(\tfrac{7}{32}, \tfrac{39}{32}\right)$ | 0.0385548104 | $2.6 \times 10^{-9}$ |
| $\left(0, \tfrac{3}{2}\right)$ | $\left(\tfrac{77}{96}, \tfrac{77}{96}\right)$ | $-0.021280658$ | $8.7 \times 10^{-9}$ |
| $\left(-2, \tfrac{3}{2}\right)$ | $\left(\tfrac{95}{32}, -\tfrac{1}{32}\right)$ | 0.0004525024 | $5.0 \times 10^{-8}$ |
| $\left(2, \tfrac{3}{2}\right)$ | $\left(-\tfrac{1}{32}, \tfrac{95}{32}\right)$ | 0.0004525024 | $5.0 \times 10^{-8}$ |
| $\left(0, \tfrac{5}{2}\right)$ | $\left(\tfrac{221}{96}, \tfrac{221}{96}\right)$ | $-0.0000356378$ | $8.8 \times 10^{-7}$ |
| $\left(-4, \tfrac{1}{2}\right)$ | $\left(\tfrac{119}{32}, \tfrac{55}{32}\right)$ | $-0.0000029746$ | $8 \times 10^{-6}$ |
| $\left(4, \tfrac{1}{2}\right)$ | $\left(\tfrac{55}{32}, \tfrac{119}{32}\right)$ | $-0.0000029746$ | $8 \times 10^{-5}$ |

Table 10: Computing the means and coefficients of variations of structure constants for the spectrum $\mathscr{S}_{2\mathbb{Z},\mathbb{Z}+\frac{1}{2}}$ at $c = 0$ with 20 choices of the points $\{z_i\}$.

[4] R. Vasseur, A. Gainutdinov, J. L. Jacobsen and H. Saleur, *Puzzle of Bulk Conformal Field Theories at Central Charge c=0*, Phys. Rev. Lett. **108**(16), 161602 (2012), doi:10.1103/PhysRevLett.108.161602.

[5] R. Vasseur, J. L. Jacobsen and H. Saleur, *Logarithmic observables in critical percolation*, J. Stat. Mech.: Th. Exp. L07001 (2012), doi:10.1088/1742-5468/2012/07/L07001.

[6] C. Fortuin and P. Kasteleyn, *On the random-cluster model*, Physica **57**(4), 536 (1972), doi:10.1016/0031-8914(72)90045-6.

[7] B. Duplantier, *Conformal Fractal Geometry and Boundary Quantum Gravity*, arXiv:math-ph/0303034; *Fractal Geometry and Applications: A Jubilee of Benoît Mandelbrot* (M. L. Lapidus and M. van Frankenhuysen, eds.), Proc. Symposia Pure Math. vol. 72, Part 2, 365-482 (AMS, Providence, R.I.) (2004), doi:10.1090/pspum/072.2/2112128.

[8] G. Delfino and J. Viti, *On three-point connectivity in two-dimensional percolation*, J. Phys. A **44**, 032001 (2011), doi:10.1088/1751-8113/44/3/032001.

[9] J. Jacobsen, *Loop models and boundary CFT*, in *Conformal Invariance: an Introduction to Loops, Interfaces and Stochastic Loewner Evolution*, Lect. Notes in Phys. **853**, 141 (2012), doi:10.1007/978-3-642-27934-8_4.

[10] G. Delfino and J. Viti, *Potts q-color field theory and scaling random cluster model*, Nucl. Phys. B **852**, 149 (2011), doi:10.1016/j.nuclphysb.2011.06.012.

[11] S. Ribault, *Conformal field theory on the plane*, arXiv:1406.4290.

[12] S. Rychkov, *EPFL Lectures on Conformal Field Theory in D≥ 3 Dimensions*, arXiv:1601.05000.

[13] S. El-Showk, M. F. Paulos, D. Poland, S. Rychkov, D. Simmons-Duffin and A. Vichi, *Solving the 3D Ising Model with the Conformal Bootstrap*, Phys. Rev. D **86**, 025022 (2012), doi:10.1103/PhysRevD.86.025022.

[14] B. Estienne and Y. Ikhlef, *Correlation functions in loop models*, arXiv:1505.00585.

[15] Al. B. Zamolodchikov, *Two-dimensional conformal symmetry and critical four-spin correlation functions in the Ashkin-Teller model*, Sov. Phys. JETP **63**, 1061 (1986).

[16] Vl. S. Dotsenko and V. A. Fateev, *Conformal algebra and multipoint correlation functions in 2D statistical models*, Nucl. Phys. B **312**, 691 (1984), doi:10.1016/0550-3213(84)90269-4.

[17] H. Saleur, *Conformal invariance for polymers and percolation*, J. Phys. A: Math. Gen. **20**(2), 455 (1987), doi:10.1088/0305-4470/20/2/031.

[18] G. Delfino, *Parafermionic excitations and critical exponents of random cluster and o (n) models*, Ann. Phys. **333**, 1 (2013), doi:10.1016/j.aop.2013.02.009.

[19] P. Di Francesco, H. Saleur and J.-B. Zuber, *Relations between the coulomb gas picture and conformal invariance of two-dimensional critical models*, J. Stat. Phys. **49**(1-2), 57 (1987), doi:10.1007/BF01009954.

[20] Vl S.. Dotsenko and V. A. Fateev, *Four-point correlation functions and the operator algebra in 2D conformal invariant theories with central charge $C \leq 1$*, Nucl. Phys. B **251**, 691 (1985), doi:10.1016/S0550-3213(85)80004-3.

[21] L. Chayes and J. Machta, *Graphical representations and cluster algorithms ii*, Physica A **254**(3), 477 (1998), doi:10.1016/S0378-4371(96)00438-4.

[22] T. M. Garoni, G. Ossola, M. Polin and A. D. Sokal, *Dynamic critical behavior of the Chayes–Machta algorithm for the random-cluster model, i. two dimensions*, J. Stat. Phys. **144**(3), 459 (2011), doi:10.1007/s10955-011-0267-y.

[23] M. Picco, R. Santachiara, J. Viti and G. Delfino, *Connectivities of Potts Fortuin-Kasteleyn clusters and time-like Liouville correlator*, Nucl. Phys. B **875**, 719 (2013), doi:10.1016/j.nuclphysb.2013.07.014.

[24] R. M. Ziff, J. J. H. Simmons and P. Kleban, *Factorization of correlations in two-dimensional percolation on the plane and torus*, J. Phys. A: Math. Gen. **44**(6), 065002 (2011), doi:10.1088/1751-8113/44/6/065002.

[25] Y. Ikhlef, J. L. Jacobsen and H. Saleur, *Three-Point Functions in $c \leq 1$ Liouville Theory and Conformal Loop Ensembles*, Phys. Rev. Lett. **116**(13), 130601 (2016), doi:10.1103/PhysRevLett.116.130601.

[26] S. Ribault and R. Santachiara, *Liouville theory with a central charge less than one*, JHEP **8**, 109 (2015), doi:10.1007/JHEP08(2015)109.

[27] J. J. H. Simmons and J. Cardy, *Twist operator correlation functions in O(n) loop models*, J. Phys. A: Math. Gen. **42**(23), 235001 (2009), doi:10.1088/1751-8113/42/23/235001.

[28] J. L. Jacobsen, P. Le Doussal, M. Picco, R. Santachiara and K. J. Wiese, *Critical Interfaces in the Random-Bond Potts Model*, Phys. Rev. Lett. **102**(7), 070601 (2009), doi:10.1103/PhysRevLett.102.070601.

[29] V. S. Dotsenko, *Correlation function of four spins in the percolation model*, Nucl. Phys. B **911**, 712 (2016), doi:10.1016/j.nuclphysb.2016.08.032.

[30] A. B. Zamolodchikov and Al. B. Zamolodchikov, *Conformal bootstrap in Liouville field theory*, Nucl. Phys. B **477**, 577 (1996), doi:10.1016/0550-3213(96)00351-3.