# Peer review of "A conformal bootstrap approach to critical percolation in two dimensions"

_SciPost Physics, doi:SciPost Phys. 1, 009 (2016)_

## Round 1 · Referee Report · Anonymous · 2016-9-20

Strengths
The problem of studying four-point functions for the random-cluster model is interesting.
Comparing numerical results of Monte Carlo and conformal bootstrap methods is also of interest.
Weaknesses
The quality of presentation is below what I would consider acceptable for publication in a journal. For example, the tables and figures have no labels or captions, which is decidedly unhelpful for the reader.
The discussion of the numerical methods and results also lacks the amount of care/rigour that would be expected for publication in a journal. This is expanded on below.
Report
This article considers the critical scaling of four-point connectivity functions of the two-dimensional Fortuin-Kasteleyn random-cluster model. Four such functions are considered. These functions are estimated via Monte Carlo simulations, using the Chayes-Machta algorithm. The main contribution is to also study four related CFT correlators, via conformal bootstrap arguments using conjectured spectra. These correlators are linear combinations of the connectivity functions of interest. The evidence presented suggests three of the four required linear combinations have been correctly identified.
Requested changes
1. Include captions and labels on all tables and figures.
2. Please put axes labels on all the plots - the plots on page 8 have no labels on the vertical axes.
3. In Section 2.2 it is stated that "10" random choices of the z_i are chosen. Why 10? To what extent are the resulting estimates robust with respect to this ad hoc choice? Some serious discussion of the numerical methodology being employed here should be given. At present the discussion is far too cavalier to give the reader serious confidence in the results.
4. Also in Section 2.2, an ad hoc choice of when to claim a spectrum is consistent vs inconsistent is given, in terms of the size of the relative error. This definition seems as reasonable as any other, but it should be stated more clearly that these ad hoc thresholds are what the authors are using to "define" consistency vs inconsistency. As presently written, the relevant paragraph seems to suggest "consistent" and "inconsistent" have inherent meanings beyond these ad hoc thresholds of relative error.
5. For the results in the table at the top of page 6, why are results only presented for N=24? To what extent are the results robust against varying the value of N? These results would be more convincing if the reader could see the N dependence for themselves.
6. At the top of page 7: How many different values of the distance parameter l are considered? This should be commented on, as should the robustness of the estimates with respect to l.
7. It is claimed that fits are performed to (3.1), but its not clear exactly what the data set is. Are multiple l values used in the fits? It seems that only one L value is considered - is that correct? If so, why? Surely to estimate parameters in a finite-size scaling ansatz it would seem prudent to consider more than one L value. How were the fits performed? Least-squares with Levenberg-Marquardt? Or some other method? Some discussion is warranted here so that the reader knows what the results actually correspond to.
8. Also related to the discussion of (3.1) it is claimed that 2 by 10^4 FK configurations are generated. Are these independent? Or are they consecutive states in the Chayes-Machta algorithm? If the latter, then due to the large autocorrelation time at the critical point of L=8192 this number of samples is actually very small. The issue of autocorrelations should be considered and commented on seriously. The claim that averaging over different z_i values in each FK configuration produces 10^12 samples likewise ignores the fact that these samples will be highly correlated, and may mislead the reader into overestimating the quality of the data set.
9. How are the fits of (3.2) performed? What is the data set? Please discuss.
10. Page 8: Why were no plots of theta>0 with q>1 shown?
11. Below the plots on page 8 it is stated that "In both approaches, the error bars are so small that it is useless to plot them". I strongly disagree with this statement. These plots should not be accepted for publication without the inclusion or error bars. If the error bars are indeed small, then the reader should be able to see this for themselves. It might be (and often is) the case that the error bars are smaller than the symbol sizes, but this is not an excuse for not including them. It is not reasonable for the reader to be expected to simply take the author's claim of small error bars at face value.
Author: Sylvain Ribault on 2016-09-20 [id 56]
(in reply to Report 1 on 2016-09-20)On points 3, 4, 5: as indicated on page 6 of the submitted article, the code for the numerical bootstrap calculations is publicly available. Readers are invited to run it, and to change the values of the parameters as they please.
Sylvain Ribault on 2016-10-03 [id 58]
There is a potentially confusing mistake on page 5, eq. (2.5), in the table that gives the ground state of $S_{2\mathbb{Z},\mathbb{Z}+\frac12}$. The ground state actually does not change as a function of $\beta$, as the diagonal state with dimension $\Delta_{(1,0)}$ does not belong to the spectrum.
The related comment on level crossing in the Conclusion should therefore be ignored, as there is no level crossing in the particular spectrum $S_{2\mathbb{Z},\mathbb{Z}+\frac12}$.

---

## Round 1 · Referee Report · Anonymous · 2016-10-4

Strengths
1- Having information about spectra and 4-pt functions of non-trivial non-rational CFTs is interesting, especially in connection with a realization in the family of q-state Potts models
2- To my knowledge, treatments of CFTs with non-diagonal spectra are scarce and the authors discuss a number of aspects in that context that I find interesting in their own right
3- The suggested mixture of analytic and numerical techniques appears to be new
Weaknesses
1- There are a few places in the article where notions are used in a way which may not be entirely standard and hence might give rise to confusion in part of the readership
2- While some of the basic foundations are explained in full detail, apparently targeted at a general audience, other aspects which are not even entirely natural for people with some but not full familiarity with the field are only mentioned in passing
3- The numerical analysis appears to be quite ad hoc at times, lacking more detailed justification
Report
The paper studies four-point functions of the q-state Potts model using a mixture of analytical and numerical techniques. Bootstrap ideas are used to motivate restrictions on the possible form of CFT spectra arising in the underlying expansion into conformal blocks. The consistency of this expansion is verified by numerical means. The resulting four-point functions are then compared with direct Monte-Carlo simulations of connectivities of random clusters, providing evidence for their coincidence.
Requested changes
-1 It is not wrong but it feels strange to see "spectrums" instead of "spectra"
-2 Page 2: With regard to the statement "a four-point function encodes (part of) the spectrum of the model, and studying its limit when two points coincide can tell us what the ground state is, whether there is a gap, etc.": Seeing the word "ground state" and "gap" in a single sentence, I have a very different image than what the authors have in mind - specifically all CFTs are regarded as gapless by certain communities. I would suggest that the authors define more carefully what they have in mind.
-3 Page 3: What is meant with "It follows that the ground state of the corresponding spectrum again has conformal dimensions..."? For me, the ground state is a property of the system as a whole and not of specific correlators.
-4 Page 5: Equation (2.1) might deserve a footnote why the usual (and more restrictive) constraints from modular invariance of the torus partition function do not play a role here.
-5 Page 4: Around (2.3) the authors assume that it is natural to have CFTs with genuinely complex conformal dimensions. In my opinion this calls for some additional explanation. Is it just a precaution to permit these? Is it an intrinsic feature of the model considered? If so, what is the physical relevance? All this might help readers who want to apply similar methods to less pathological examples to understand what is going on.
-6 While "spectrum" may be a matter of choice, the plural of "ansatz" is quite certainly not "ansatzes" but "ans\"atze"
-7 Page 5: Below (2.1) the authors write "In particular, states such that ... are called diagonal, and spectrums that only involve such states are called diagonal too." Part of this statement only makes sense when speaking about "highest weight states" or "ground states" (of highest weight modules) and the authors should adjust this correspondingly

---

## Round 2 · Author Response

We are grateful to the first referee for his technical remarks, and to the second referee for his insightful suggestions that led us to perform significant clarifications.

---

## Round 2 · List of Changes

Replies to the first referee report:

  1. Our way of inserting tables and figures in the text seems better to us. We will not change it unless given a good reason to do so.

    1. and 5. We have added in the appendix outputs of the numerical conformal boostrap obtained by varying the number of states and the number of drawing of the sample points. We added a sentence in the last paragraph of page 5 giving more details of how we set precision of the conformal boostrap.
  2. and 7. We considered 47 values of l's while taking 4 corrections b_1..b_4 and c_1..c_4. We also considered smaller lattice size with similar fits and checked that finite size corrections are negligible for L=8192. For each fit, the quality is very high (reduced chi-square much lower than one). The fit is done with a least-squares Levenberg-Marquardt algorithm. All this is specified in the new version.

  3. We had written 2 X 10^4 randomly generated graphs which is meaning 2 X 10^4 independent configurations. At the time of writing the letter, we had if fact 10^5 configurations for most values of Q except for some values for which there was only 2 X 10^4 configurations. We have now 10^5 configurations for all values of Q so 2 X 10^4 is replaced by 10^5 in the text. We also replaced "randomly generated graphs" by "independent configurations"

Concerning the 10^13, we wrote explicitly N L^2 \simeq 10^13 and it was clearly stated that L^2 is the number of measurements for each graph (ie configuration). We do not see how this can mislead a reader ?

  1. and 11. We include figures with the difference of the two types of data (Monte Carlo and Bootstrap) with error bars. The fit is obtained by minimizing the differences.

  2. We checked in details only for the case q=1 (ie percolation) in the whole complex plane since this is the fastest case. Doing the same for large values of q would have taken a lot of computational resources. We do not believe that we would have gained much more information.

Replies to the second referee report:

  1. & 6. The regular plurals 'spectrums' and 'ansatzes' can be found in dictionaries and encyclopedias, alongside the more common but irregular forms 'spectra' and 'ansätze'. Life would be easier, especially for non-Western scientists, if Latin and German plurals were eliminated whenever possible. Following the referee's advice, we however reverted to 'ansätze'.

  2. We have replaced the notion of having a gap with the distinction between continuous and discrete spectrums.

  3. We replaced the expression 'ground state' with 'leading state' (of a given spectrum), and defined it when it first occurs.

  4. We have added a comment on modular invariance at the end of Section 1.

  5. We have added a comment on unitarity after (2.4). We see no other reason why conformal dimensions might be real. But the theories we are considering must be non-unitary for generic values of c.

  6. We corrected the definition of diagonal and even spin spectrums, by writing them as conditions on primary states.

Other changes:

  1. We deleted the assumption |\beta| \leq 1 from (1.1), because we are not sure that it is the right assumption when q is not real. Instead, we added the assumption \frac12 \leq \beta^2 \leq 1 after (1.5).

  2. We deleted the distinction between the cases |\beta|^2 \leq \frac12 and |\beta|^2 \geq \frac12 in (2.5), as it was the result of a mistake.

  3. We added references [16]-[19] at the end of Section 2.1.

  4. We added a sentence at the end of Section 3.3 to point out analogies of our interpretation with previous works .

---

## Editorial Decision

published